# The Ability of Large Language Models to Evaluate Constraint-satisfaction in Agent Responses to Open-ended Requests

## Abstract

Generative AI agents are often expected to respond to complex user requests that have No One Right Answer (NORA), e.g., *design a vegetarian meal plan below 1800 calories*. Such requests may entail a set of *constraints* that the agent should adhere to. To successfully develop agents for NORA scenarios, an accurate automatic evaluation framework is essential, and specifically - one capable of validating the satisfaction of constraints in the agent's response. Recently, large language models (LLMs) have been adopted as versatile evaluators for many NORA tasks, but their ability to evaluate constraint-satisfaction in generated text remains unclear. To study this, we develop and release a novel Arithmetic Constraint-Satisfaction (ACS) benchmarking dataset. The dataset consists of complex user requests with corresponding constraints, agent responses and human labels indicating each constraint's satisfaction level in the response. A unique property of this dataset is that validating many of its constraints requires reviewing the response as a whole (in contrast to many other benchmarks that require the validation of a single independent item). Moreover, it assesses LLMs in performing reasoning, in-context data extraction, arithmetic calculations, and counting. We then benchmark both open and proprietary LLMs on evaluating constraint-satisfaction, and show that most models still have a significant headroom for improvement, and that errors primarily stem from reasoning issues. In addition, most models exhibit a skewed constraint-satisfaction prediction pattern, with higher accuracy where the ground-truth label is *satisfied*. Lastly, few-shot prompting for our task proved to be rather challenging, since many of the studied models showed a degradation in performance when it was introduced.

## 1 Introduction

Generative AI agents are becoming increasingly popular, especially with the development of large language models (LLMs). As these models become more advanced and autonomous, the scope of AI agents increases, and they are now designed to include advanced capabilities and skills, such as numerical reasoning, planning, and using external tools (Pan et al., 2023; Di Palo et al., 2023; Wang et al., 2024; Qin et al., 2023). Powered by these capabilities, they are expected to handle complex user requests that may require the agent to perform multiple steps, and adhere to constraints that may be imposed by the request. Examples for such requests include planning a trip with a given budget, creating a meal-plan with specific daily caloric-intake, or generating a fictional story with a specific number of acts and characters. To facilitate the development of AI agents capable of addressing such complex requests, evaluating the quality of the agent's response is essential. To illustrate this concept, the top part of Figure 1 shows an example of a complex user request to an AI agent that uses external tools, reasoning, and multi-step planning to provide an adequate response, which is finally evaluated to understand how well the response addressed the user request.

Evaluating agent responses to complex user requests is a challenging task, especially for requests that have No One Right Answer (NORA). To alleviate this, we suggest to focus on a subset of NORA requests that correspond to a Well-defined, Objective, and Verifiable (WOV) evaluation criteria, e.g., "design a *3-day* meal-plan with *no meat* products". Evaluating whether the agent's response provides 3 days of a meal plan and does not include meat is a feasible, well-defined and objective task. This is

in contrast to other NORA requests that correspond to a *subjective* evaluation criterion, e.g., "write a *funny* song about cats" (evaluation of *funny* is highly subjective and may depend on time, location, and culture), or requests include *fuzzy* or *relative* evaluation criteria, e.g., "plan a *short* trip to Paris with a *small* budget". It should be noted that some requests may only partially correspond to WOV evaluation criteria, i.e., not all parts of the request can be objectively evaluated. Nonetheless, the ability to evaluate an agent's response with respect to only some parts of the request can be useful as well. Moreover, such requests are diverse in the sense that they span many domains and use-cases. Therefore, an evaluation framework for these cases can be very useful for developing AI agents.

To evaluate agent responses in the scenarios described above, we propose a *constraint-satisfaction* framework/protocol. In this protocol, a set of *constraints* is extracted from the user request, followed by an assessment of the constraint-satisfaction level in the agent's response for each constraint in the set. Thus, the evaluation criterion is the alignment between the agent's response and the constraints that are imposed by the user request. This is illustrated in Figure 1 where a user asks for a trip plan with many constraints entailed in the request. The constraints are enumerated and a set of constraints is produced. Then, the agent's response is evaluated against each constraint iteratively and independently. A final score can then be given. Note that the constraints are formulated in natural language, and thus, the scope of evaluation is not limited by this protocol. In addition, such a protocol was previously studied and was found useful for detecting factual errors in LLM' responses using attention patterns (Yuksekgonul et al., 2023) and for information-retrieval (Abdin et al., 2023).

Evaluating the constraint-satisfaction level in agent responses can be performed in multiple ways. One option is to utilize human raters, but this approach is often not reproducible, and more importantly, it is not scalable. Thus, an automatic evaluation framework is highly desired. Recently, many works have utilized LLMs for various evaluation tasks, especially when the evaluation criterion is becoming more complex and intricate, such as in NORA scenarios (Chang et al., 2024; Li et al., 2024; Zheng et al., 2024; Liu et al., 2023a). For example, (Xu et al., 2023; Kasahara & Kawahara, 2023; Chan et al., 2023; Qin et al., 2023; Wang et al., 2023) studied LLMs as side-by-side evaluators, (Fu et al., 2023; Chen et al., 2023; Lin & Chen, 2023; Liu et al., 2023b; Zhong et al., 2022; Chiang & Lee, 2023) studied LLMs for evaluating a pre-defined set of attributes (e.g., accuracy, coherence, and informativeness), and (Jiang et al., 2023; Min et al., 2023; Lu et al., 2023) studied more advanced evaluation protocols based on error analysis. However, they did not explicitly study the ability of LLMs in evaluating constraint-satisfaction in NORA scenarios. In order to enable this, a specific benchmarking dataset is required.

There are many datasets that enable benchmarking LLMs on separate capabilities that are required for evaluating constraint-satisfaction. For instance, (Cobbe et al., 2021; Patel et al., 2021; Roy & Roth, 2016) test arithmetic reasoning, (Rajpurkar, 2016; Joshi et al., 2017) test question-answering, (Hendrycks et al., 2020) tests multi-level knowledge in a diverse set of fields, and (Thakur et al., 2021) test information-retrieval. While providing useful insights into diverse LLM capabilities, they do not test LLMs in NORA scenarios. Thus, understanding the performance of state-of-the-art LLMs in evaluating constraint-satisfaction is still somewhat limited.

To fill this gap, in this work we develop a dataset for benchmarking LLMs on the task of evaluating constraint-satisfaction in NORA scenarios. The dataset is semi-structured: each datapoint comprises a user request, a constraint, an agent response, and a binary label (annotated by human raters) for whether the constraint is satisfied, all formulated in natural language. We chose to focus on numerical constraints in order for the task to be well-defined, and since their evaluation require complex multi-step reasoning over the entire agent's response. With this configuration, the benchmark assesses the LLM's ability to perform multiple steps in-context to arrive at the final answer, where each step may require a different capability: reasoning, data extraction, arithmetic calculations, and counting. We thus name the benchmark ACS for *Arithmetic Constraint-Satisfaction*. We use the dataset to benchmark both proprietary and open popular LLMs. The contributions of our work are as follows.

1. Formulation of a *constraint-satisfaction framework* that facilitates automatic evaluation of agent responses to complex user-requests in NORA scenarios.

2. Development and release of the ACS dataset for benchmarking auto-scorers of *constraint-satisfaction*. The dataset if available at *blinded for submission*.

3. Benchmarking current state-of-the-art (SOTA) LLMs on the ACS dataset, including both proprietary models (Gemini 1.5, GPT-4o), and open models (Llama-3, Mixtral, Mistral),

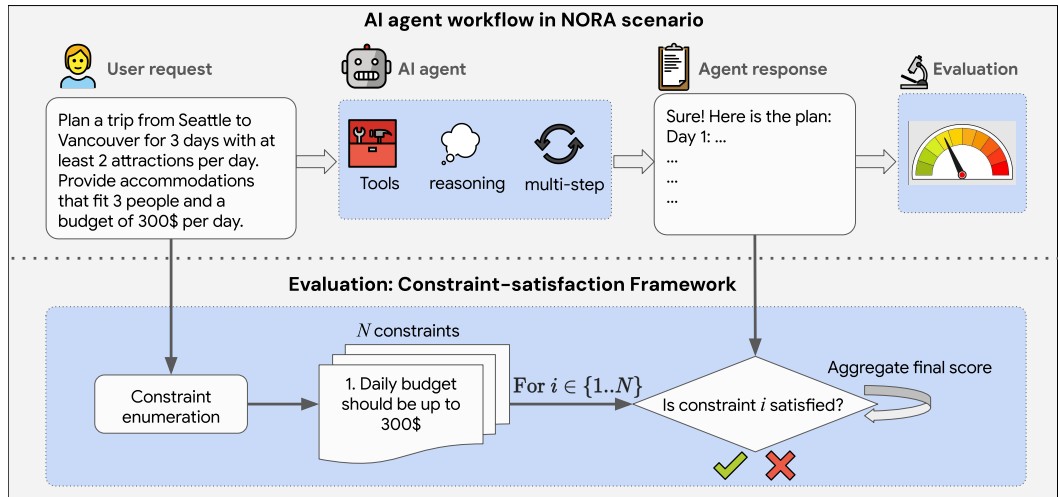

Figure 1: An illustration of a complex user request to an AI agent for planning a trip with constraints. The agent should typically use reasoning, external tools and take multiple steps to provide an adequate response. Then, an evaluation process should be performed to score the quality of the response. At the bottom part, the constraint-satisfaction protocol is illustrated, where, first a set of constraints that should be satisfied in the agent's response is enumerated from the user request. Then, the evaluation process assesses the constraint-satisfaction level in the response iteratively, for each constraint in the set.

revealing weaknesses in some models' capability to serve as auto-raters, as well as the challenges in effectively implementing few-shot prompting.

4. Follow-up error analysis showing that *reasoning* is the main cause of error, and not arithmetic calculations.

## 2 ARITHMETIC CONSTRAINT-SATISFACTION DATASET

This section describes the ACS benchmarking dataset, including its development, properties, scope and limitations. The aim of the ACS dataset is to benchmark LLMs on the task of *evaluating constraint-satisfaction in NORA scenarios*. More specifically, the dataset is focused on WOV constraints that require the LLM to perform reasoning, data extraction, arithmetic calculations, and counting. The scenarios in the dataset are taken from three domains of high interest: meal-planning, daily-schedule-planning, and workout-planning. At a high-level, each datapoint in the dataset is structured as follows, where each item is formulated in natural language:

- **User request**: a NORA request to an AI agent that contains at least one WOV constraint.

- **Constraint**: a single constraint that corresponds to the user request that should be verified in the agent's response.

- **Agent response**: a generated response that addresses the user request.

- **Label**: a human-annotated binary label for whether the constraint is satisfied or not.

Note that the constraints that should be verified in the response are given explicitly in the dataset, thus the scoring LLM is not required to extract these from the user request. The reason for this is to provide a common evaluation criterion to different LLMs (i.e., how well do different LLMs evaluate the constraint-satisfaction level of a *specific* constraint). However, future work could study and compare LLMs' performance in evaluating the satisfaction of implicit constraints. In addition, note that each datapoint corresponds to a single constraint, even though there may be multiple constraints to each request. This structure facilitates the performance analysis of different LLMs on the constraint-level, rather then request-level.

## 2.1 DATASET GENERATION PROCESS

The dataset was generated using an interleaved process of LLM prompting for generating text (user request, constraints, and agent responses), and manual modifications and filtering of the generated text (performed by humans). The latter was performed to refine the LLM output and fix inconsistencies. We used Gemini-1.0-ultra (Gemini Team Google, 2023) to generate the text in all of the stages describes next:

1. [Manual] Crafting guidelines for how to generate user requests that entail complex arithmetic constraints. These can be thought of as seed prompts for generating the entire dataset. We used four sets of guidelines, one for each domain: meal-planning, and daily-schedule-planning, and we further separated the workout-planning domain into cardio and strength, and provided a different set of guidelines for each sub-domain. The guideline in each domain included specific constraints that should be explicitly stated in the user request. For example, in the meal-planning domain, the guideline included a caloric restriction value that should be requested, with a value taken from a reasonable pre-defined appropriate range.

2. [Gemini-1.0-ultra] Generating user-requests in the three domains according to the manually crafted guidelines in the previous step.

3. [Manual] Appending a final instruction to each user request that was generated in the previous step. The final instruction requested to explicitly include a breakdown of relevant numerical information, e.g., number of calories in meal-plan, working time in a daily-schedule, and exercise duration in cardio-workout-plan. This step was taken to verify that the generated responses to the user queries would explicitly address the constraints and would include numerical values that could be later evaluated by an auto-scoring system.

4. [Gemini-1.0-ultra] Generating constraints for each user request that were created in step 3. In this step, Gemini was given few-shot examples in order to generate only arithmetic and counting related constraints and to control the format of the constraints.

5. [Manual] Correcting the format or phrasing of the generated constraints in the previous step or adding missing constraints.

6. [Gemini-1.0-ultra] Generating "agent responses" by querying Gemini with the user requests that were created in step 3. While using Gemini-1.0-ultra, i.e., an LLM, rather than a more advanced AI agent with planning capabilities may seem inappropriate at first, recall that the aim of the ACS dataset is to benchmark LLMs on evaluating constraint-satisfaction. Thus, the agent responses in the dataset are not required to be generated by a domain-dedicated AI agent.

7. [Manual] Filtering and modifications to the agent responses that were generated in the previous step, e.g., removing information from the response that may cause the constraint-satisfaction evaluation to be ambiguous, or revising the response to control whether the constraint is satisfied or not in order to diversify the data.

8. [Manual] Labeling each pair of *constraint* and *agent response* as either *satisfied* or *unsatisfied*.

With the process above, 25 unique user requests were generated for each domain: meal-planning, daily-schedule-planning, workout-planning-cardio, and workout-planning-strength. Each request corresponds to multiple constraints and to a single agent response. Next, we have separated the constraints such that each datapoint in the dataset would correspond to a unique triplet of {user request, constraint, agent response}. Following all the steps above (which included manual removal and augmentation of the data) resulted in a dataset with 405 datapoints, and with a satisfied to unsatisfied datapoints ratio of 241/164 ($\approx$59% of samples are labeled as *satisfied*). An example for a datapoint is presented in Figure 2.

## 2.2 DATASET EVALUATION SCOPE

The agent responses in the ACS dataset may include inaccuracies and inconsistencies that are not expected to be validated by the scoring LLM, and their verification is out of scope of this work. For instance, a meal-plan may include a food item with a corresponding caloric value, but with no exact measurements. On the other hand, it may include exact measurements, but the given caloric value

| User request: Can you design a 2-day meal plan for me with 3 meals each day? The daily caloric intake should be 2900 calories, with at least 150 grams of protein and a maximum of 90 grams of fat. I'm a pescetarian, so no meat, please. Make sure to include a breakdown of calories, protein, and fat for each food item in the plan. | Constraint: The total daily protein intake must be at least 150 grams | Agent response: **2-Day Pescetarian Meal Plan (2900 Calories)**

**Day 1**

**Breakfast**
* 1 cup oatmeal with 1/4 cup blueberries and 1/4 cup chopped walnuts (450 cal, 15g protein, 15g fat)
* 2 slices whole-wheat toast with 1 tbsp almond butter (250 cal, 10g protein, 10g fat)
* 1/2 cup Greek yogurt (100 cal, 12g protein, 0g fat)

**Lunch**
* 4 oz grilled salmon with 1 cup brown rice and 1 cup broccoli (500 cal, 40g protein, 20g fat)
* 1/2 cup lentil soup (200 cal, 15g protein, 5g fat)
* 1 apple with 1 tbsp peanut butter (100 cal, 5g protein, 5g fat)
…
… | Label: Unsatisfied |

Figure 2: A datapoint example from the ACS dataset. The full agent response was trimmed for brevity. In this case, the constraint is unsatisfied since Day 1 corresponds to a total protein intake that is less then 150 grams.

may be incorrect. Another example in the workout-planning domain is a cardio routine with a given value for estimated burned calories, which may be very inaccurate. Evaluating this level of accuracy in the agent response is not the aim of the ACS benchmark. Rather, the numerical values that are given in the response, which can not be further broken down into smaller components (based on the information that is given in the response), are assumed to be correct. However, any global numerical values, such as a response that states "Here is a meal plan with at least 150 grams of protein each day.." are not assumed to be automatically correct, and should be verified by the scoring LLM. Thus, most of the datapoints in the ACS dataset can be evaluated using the following general process:

1. Extract information from the agent's response that is relevant to the current constraint, either numerical values (such as caloric values for each food item in a given day) or other textual entities (such as a list of exercises that comprise a single routine).

2. Perform arithmetic operations (such as summation, multiplication, or subtraction), or counting.

3. Evaluate the result with respect to the constraint.

4. Potentially repeat the steps above with a different section in the agent's response (for instance, evaluating the caloric intake of the next day in the meal plan).

## 2.3 REQUIRED NUMERICAL CAPABILITIES

The specific numerical capabilities that are required to evaluate each datapoint in the dataset are: counting, summation, multiplication, and date-time arithmetic. The distribution of the required capabilities in each datapoint in the ACS dataset is presented in Figure 3. Date-time arithmetic mainly refers to the ability to understand how much time is assigned to different sections in a given schedule. Concretely, this means calculating the duration between two specific times within a given schedule, where the times are mostly expressed in "HH:MM" format. In some datapoints, the total time should be accumulated based on multiple sections in the schedule. All the datapoints that require performing *multiplication* also require accumulating the results over multiple sections in the response. Thus, this capability is explicitly stated as "Multiplication and summation" in Figure 3.

## 2.4 DATASET PROPERTIES

To make this benchmark realistic and challenging, it was designed to have several key properties. First, the relevant information that the LLM should use in its evaluation is not presented sequentially, but is rather scattered in the context window. The maximal number of tokens from the *agent response* field in the ACS dataset is 1963, when calculated with Gemini 1.5 Pro tokenizer via the Gemini API

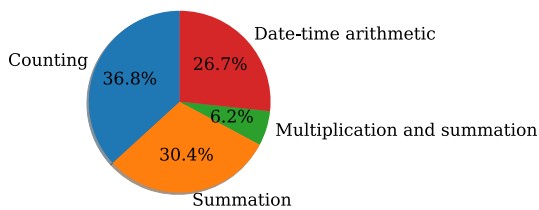

Figure 3: The distribution of the required numerical capabilities in each datapoint in the ACS dataset. Datapoints that require multiplication always require summation as well, and thus "Multiplication and summation" is stated explicitly.

(Google, 2024). This is far smaller than the maximal context size of current state-of-the-art LLMs, many of which support 8k to 2M input tokens (including those that will be studied in Section 3). Another property of the benchmark is that it may contain "distractors" for specific constraints, i.e., similar pieces of information that should be ignored since they are not relevant. Moreover, "positive distractors" may be present, i.e., keywords that represent the constraint as being satisfied (e.g., "Here is a 2000 calorie meal plan" when the constraint is "the meal-plan should be up to 2000 calories a day" and the label is "unsatisfied"). Another challenging property of our benchmark is that in order to fully verify some constraints, the LLM should perform an iterative evaluation process. This refers to performing multiple independent instances of the same evaluation process using different pieces of information from the context. An example for this is when evaluating the daily caloric intake of a multi-day meal plan, the calories for each day should be calculated independently. Thus, unlike benchmarks that rely on simple keyword matching or isolated text snippets (such as in Question-Answering, NLI, and sentiment analysis), the ACS dataset demands a comprehensive/holistic evaluation of the agent response. Lastly, the benchmark does not require any domain-specific or specialized knowledge. The complexity of evaluating each datapoint in the dataset can be considered to be at an elementary-school level. This is important in order to fairly evaluate fundamental LLM capabilities that are desirable across domains, without giving an advantage to domain-specific LLMs. We believe that these properties make the ACS benchmark useful in assessing the capabilities that are crucial to successfully incorporate LLMs into a wide range of applications.

## 3 EXPERIMENTS

The ACS dataset was used to benchmark several LLMs on the task of *evaluating arithmetic constraint-satisfaction* in NORA scenarios. Recall that the ACS dataset contains a ground-truth binary label for whether the constraint is satisfied in the agent's response. Thus, in this study, the LLMs were instructed to evaluate the agent's response with respect to the constraint, and were instructed to provide a final *yes/no* decision for whether the constraint is satisfied.

### 3.1 SETUP

The studied LLMs include Gemini 1.5 Pro (version 0514), Gemini 1.5 Flash (version 0514), Gemini 1.0 pro (stable version 002) (Gemini Team Google, 2023), GPT-4o (version 2024-05-13) (OpenAI, 2024), Llama-3-70b-chat, Llama-3-8b-chat (AI@Meta, 2024), Mixtral-8x7b-instruct-v0.1 (Jiang et al., 2024), and Mistral-7b-instruct-v0.2 (Mistral AI, 2024). Gemini models were accessed through the Gemini API, OpenAI's GPT-4o was accessed through the OpenAI API, and the open models were deployed to a Vertex AI endpoint. Default text-generation parameters were used for each LLM. Each LLM was then used to evaluate the entire ACS dataset. The LLMs were instructed to use a chain-of-thought (CoT) reasoning process (Wei et al., 2022) and perform any necessary calculations explicitly, rather than relying on final values stated in the given plan. Furthermore, two prompting configurations were studied: zero-shot and few-shot with two evaluation examples taken from a trip-plan scenario (which is out-of-domain). The first example contains a 3-day itinerary including prices for each element and a daily-budget constraint. The second contains a driving plan of multiple segments with driving time and average speed for each segment and a constraint of

maximal driving distance per segment. The evaluation process in the example shows the model how to extract the relevant information (item prices, or driving time and average speed), perform calculations (summation and multiplication), and compare the result against the constraint to decide whether it is satisfied. The evaluation prompt is presented in the appendix in section A.1, and the few-shot examples are presented in section A.2. Then, each LLM's final decision, i.e., prediction, regarding the constraint-satisfaction in the agent's response was extracted from the full LLM evaluation response using regex, and was used to analyze the performance of the LLM.

## 3.2 RESULTS - ACCURACY METRICS

Following the predictions that were made by each LLM, the following accuracy metrics were calculated: overall-accuracy, which is the constraint-level prediction accuracy, and $F_1$ score of predicting each of the labels: "satisfied" and "unsatisfied". By this separation, we can study whether an LLM has a bias towards predicting one label over the other. The accuracy metrics are presented in Table 1 for all studied LLMs. As can be seen, GPT-4o achieves the best accuracy scores in both zero-shot and 2-shot configurations. The high accuracy score of 97.04% shows that GPT-4o can serve as a rather reliable auto-scorer for the kind of tasks presented in the ACS benchmark. The next best performing model is Llama-3-70b-chat at a zero-shot configuration, but it performs significantly worse, at an accuracy rate of 90.62%. The performance of the remaining models is even worse, indicating their lack of competence in performing as reliable auto-scorers in the constraint-satisfaction task studied here.

Another insight derived from Table 1 is the difference in $F_1$ scores between constraints with *positive* (satisfied) and *negative* (unsatisfied) labels. All models but GPT-4o seem to predict positive datapoints much more accurately than negative datapoints. This phenomenon may be attributed to the fact that the ACS dataset contains "positive" distractors (see subsection 2.4), i.e., keywords in the agent response that imply that the constraint is satisfied (e.g., "Here is a 2000 calorie meal plan" when the constraint is "the meal-plan should be up to 2000 calories a day" and the label is "unsatisfied"). If these "positive" distractors are indeed the causes for the imbalance between the labeling classes, it may show a weak-point of the models in performing similar tasks objectively. However, this is just a hypothesis at this stage, and further analysis is required to validate it, and is suggested for future work.

Finally, when comparing the performance of the models in the zero-shot versus 2-shot configuration as seen in Table 1, an interesting behavior is observed. Some models present an improved performance with respect to the $F_1$ score in the 2-shot configuration, such as Gemini 1.5 Flash (increase of 4.44 percentage points), Gemini 1.0 Pro (increase of 3.46 percentage points), and perhaps Gemini 1.5 Pro although the difference is not major (increase of 0.98 percentage points). This shows that these models can benefit from few-shot prompting strategies since they can guide the models to evaluate constraint-satisfaction more accurately. In contrast, the accuracy of the open models decrease in the 2-shot configuration, compared to zero-shot. This is most noticeable in Mixtral-8x7b-instruct-v0.1 (decrease of 7.16 percentage points) and Llama-3-8b-chat (decrease of 4.44 percentage points), while the remaining models correspond to a decrease of less than 2 percentage points. The cause for this decrease in performance is not clear from this analysis alone, but a potential cause may be the out-of-domain examples. Lastly, GPT-4o achieves a high level of accuracy in both zero-shot and 2-shot configurations, which is a desirable behavior that suggests increased generalization capabilities, compared to the remaining models.

## 3.3 RESULTS - ERROR ANALYSIS

To further analyze some of the LLMs' performance and gain insights into their capabilities, an error analysis of selected models was performed in the 2-shot configuration. The inspected models are Gemini 1.5 pro, Gemini 1.5 flash, GPT-4o, and Llama-3-70b-chat. Since the evaluation prompt template invokes CoT reasoning (see Appendix A.1), it enables examining the full evaluation process of each LLM and identifying the cause of error in cases of incorrect final prediction. The errors were manually analyzed and categorized into the following buckets:

1. **Reasoning**: which was further divided into these subcategories

    (a) Extraction: failing to extract all relevant items, or extracting additional irrelevant items

Table 1: Overall-accuracy, satisfied $F_1$, and unsatisfied $F_1$ scores achieved by each LLM in evaluating the constraint-satisfaction level of the ACS dataset in zero-shot and 2-shot configurations. The best-performing results are highlighted in bold.

| Model | zero-shot | | | 2-shot | | |
|---|---|---|---|---|---|---|
| | Accuracy | Satisfied $F_1$ | Unsatisfied $F_1$ | Accuracy | Satisfied $F_1$ | Unsatisfied $F_1$ |
| Gemini 1.5 Pro | 88.40% | 90.43% | 85.27% | 89.38% | 91.35% | 86.26% |
| Gemini 1.5 Flash | 84.20% | 86.44% | 81.07% | 88.64% | 90.53% | 85.8% |
| Gemini 1.0 Pro | 75.80% | 79.58% | 70.3% | 79.26% | 82.5% | 74.55% |
| GPT-4o | **97.04%** | **97.54%** | **96.27%** | **97.04%** | **97.55%** | **96.25%** |
| Llama-3-70b-chat | 90.62% | 91.95% | 88.76% | 88.64% | 90.61% | 85.62% |
| Llama-3-8b-chat | 80.49% | 82.41% | 78.12% | 76.05% | 80.00% | 70.15% |
| Mixtral-8x7b-instruct-v0.1 | 72.84% | 77.18% | 66.46% | 65.68% | 71.22% | 57.49% |
| Mistral-7b-instruct-v0.2 | 68.15% | 71.14% | 64.46% | 67.90% | 73.68% | 58.86% |

    (b) Counting: extracting a correct list of items to count but the final value is wrong

    (c) Schedule understanding: incorrect deductions from a typical schedule structure

    (d) Other: intermediate or final reasoning steps with logical errors

2. **Calculation**: errors in summation, multiplication, or date-time related errors (mainly time calculation "subtraction" errors)

Table 2 shows the number of times each error category occurred for incorrectly predicted data points. The percentage of each error category relative to the total number of errors is shown in brackets. Note that for all LLMs, most errors are caused by erroneous reasoning steps. This highlights the fact that incorporating tool-use for arithmetic calculations is not expected to be the most important step for improving the performance of these models in similar tasks. In addition, GPT-4o did not make calculation errors, but both Gemini models and Llama-3-70b-chat made such errors with relatively similar proportions. Moreover, it seems that correctly extracting in-context relevant information is more challenging for Gemini 1.5 flash, compared to the other LLMs. For Llama-3-70b-chat, counting seems to be more challenging compared to the other models. Finally, recall that a model might make any error and still predict the final label correctly. Thus, this analysis has limitations and should not be interpreted as showing that any LLM was immune to making specific errors.

Table 2: Error analysis counts of some of the studied LLMs. Absolute counts are shown and their portion from the total number of errors is in brackets. Most errors are caused by incorrect reasoning steps.

| Model | Total errors | Reasoning | | | | Calculation |
|---|---|---|---|---|---|---|
| | | Extraction | Counting | Schedule understanding | Other | |
| Gemini 1.5 Pro | 40 | 7 (17.5%) | 1 (2.5%) | 2 (5.0%) | 20 (50.0%) | 10 (25%) |
| Gemini 1.5 Flash | 48 | 16 (33.3%) | 0 | 3 (6.3%) | 18 (37.5%) | 11 (22.9%) |
| GPT-4o | 14 | 5 (35.7%) | 1 (7.1%) | 1 (7.1%) | 7 (50.0%) | 0 |
| Llama-3-70b-chat | 45 | 8 (17.8%) | 7 (15.6%) | 0 | 19 (42.2%) | 11 (24.5%) |

## 4  LIMITATIONS

The work presented here offers useful insights in to the ability of LLM to serve as auto-scorers for the task of constraint-satisfaction in NORA scenarios, but it has some limitations. First, the ACS dataset has a limited scope and size. It spans three main planning domains, which are useful, but represent only a small fraction of real-life use-cases. The set of capabilities that is required to correctly evaluate the dataset is also limited, as described in section 2, and thus it does not reflect the full set of capabilities that are required from auto-scorers to evaluate constraint-satisfaction. In addition, the size of the data is limited to 405 datapoints, where each corresponds to less than 2000 tokens. With the increased interest in very large context sizes (Lin et al., 2024; Song et al., 2024; Ding et al., 2024; Gemini Team Google, 2023), it may be very useful to study the ability of LLMs to evaluate constraint-satisfaction with very large context sizes. Currently, the ACS dataset does

not include datapoints that are composed of very large tokens ($> 10k$), but this is suggested for future work. Finally, while GPT-4o achieves very high accuracy scores when benchmarked against the ACS dataset, the remaining LLMs, especially the open models, have a significant headroom for improvement. Thus, the ACS dataset can be useful for the development of more advanced LLM-based auto-scorers.

Next, the experimental study in section 3 also has some limitations. Recall that the study in section 3.2 measures the accuracy of the LLMs in predicting the correct constraint-satisfaction label - either *satisfied* or *unsatisfied*. With the reduction of the evaluation task to *binary prediction*, it is possible for the LLM to make some errors, whether in the data-extraction, reasoning, calculation, or counting step, but nonetheless predict the correct class, assuming such potential errors are insignificant. As an example, consider the constraint: "the meal-plan should be at least 1700 calories" that corresponds to a meal-plan with 1800 calories. An LLM that during the evaluation calculates either 1800 (correct) or 2000 (incorrect) calories, could predict the same correct label: *satisfied*. This paradigm facilitates the analysis of the LLM accuracy (no additional steps are required to extract intermediate numerical values that the LLM is expected to produce) but it may obscure the LLM actual performance to some extent. For future work, additional experiments could be performed that test the LLMs accuracy in more detail, for instance, by examining the accuracy of the *numerical values* that the LLMs produce during their evaluation. In the example above, an additional step that extracts the meal-plan calories that were explicitly calculated by the LLM and verifies this value against 1800 could be very insightful.

Finally, the study in section 3.3 presents an error analysis with specific error categories, which are just a single way to cluster the error "buckets". Furthermore, the LLM could potentially make multiple errors corresponding to multiple categories, but we chose a single category that best describes the most significant error in the LLM output. Thus, this is a subjective and not an exhaustive analysis. We leave a more detailed analysis for future work.

## 5 CONCLUSIONS

This work presented a novel dataset for benchmarking auto-scorers using a constraint-satisfaction framework. The experiment results showed that the task of evaluating agent responses with respect to constraints that require performing in-context data extraction, reasoning, and elementary-school level arithmetic calculations and counting is still challenging for many state-of-the-art LLMs. GPT-4o was the only model that achieved satisfactory accuracy scores, among all the tested models. In addition, "positive distractors" in the agent's response, i.e., keywords that represent the constraint as being satisfied even though this may not be the case, may pose a challenge to LLMs that aim to score the response. Regarding the prompting strategy, not all models may benefit from few-shot prompting for the task studied by the ACS dataset. Moreover, it may be detrimental to their performance, and thus, this prompting technique should be handled with care when designing an LLM-based auto-scorer. Finally, the primary source of errors is identified to be due to *reasoning* rather than *arithmetic calculation* issues, suggesting that incorporating external tools for calculation purposes may not lead to significant performance improvements.

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

## A APPENDIX

### A.1 EVALUATION PROMPTS

The evaluation prompt that was used for all the LLMs in the experiments is shown below in Jinja syntax Pallets (2024):

```
You are required to evaluate whether an agent's response satisfies
    a given constraint.
Make sure to follow these guidelines:
1. You should be skeptical of the agent's response. Prefer to do
    your own calculations rather than relying on the values given
    in the response.
2. Your final decision should be "yes" or "no".
3. Provide the following in your response:
RATIONALE: Explain why you decided whether or not the agent's
    response satisfies the constraint. Include here any
    calculations that are required to perform the evaluation. End
    this with "# [END_RATIONALE]".
FINAL_ANSWER: Your final answer whether the constraint is fully
    satisfied in the agent's response - "yes" or "no".

{% if examples %}
Here are examples that can help you to understand how to evaluate
    different agent responses with corresponding constraints:
{% for ex in examples %}
The agent's response is:
[BEGIN AGENT RESPONSE]
{{ ex.agent_response }}
[END AGENT RESPONSE]

The constraint is: {{ ex.constraint_value }}

[BEGIN EVALUATION PROCESS]
RATIONALE: {{ ex.rationale }} # [END_RATIONALE]
FINAL_ANSWER: {{ ex.final_answer }}
[END EVALUATION PROCESS]

{% endfor %}
[END EXAMPLES]
{%endif%}

Begin! Think step-by-step before providing your response!
The agent's response is:
[BEGIN AGENT RESPONSE]
{{ agent_response }}
[END AGENT RESPONSE]

The constraint is: {{ constraint_value }}

[BEGIN EVALUATION PROCESS]
```

### A.2 EXAMPLE GIVEN TO THE LLMS FOR HOW TO EVALUATE AN AGENT'S RESPONSE

The trip-planning examples that were given to the LLMs in the 2-shot prompting configuration for how to evaluate an agent's response are shown below:

```
Here are examples that can help you to understand how to evaluate
    different agent responses with corresponding constraints:
```

The agent's response is:
[BEGIN AGENT RESPONSE]
**Day 1**

* **Breakfast:** Portage Bay Cafe ($20)
* **Attraction 1:** Space Needle ($35)
* **Attraction 2:** Museum of Pop Culture (MoPOP) ($30)
* **Lunch:** Pike Place Chowder ($20)
* **Attraction 3:** Pike Place Market ($0)
* **Attraction 4:** Seattle Waterfront ($0)
* **Dinner:** The Pink Door ($45)

**Day 2**

* **Breakfast:** Biscuit Bitch ($20)
* **Attraction 1:** Ferry to Bainbridge Island ($15)
* **Attraction 2:** Bloedel Reserve ($20)
* **Lunch:** Doc's Marina Grill ($30)
* **Attraction 3:** Seattle Art Museum ($30)
* **Attraction 4:** Olympic Sculpture Park ($0)
* **Dinner:** Lola ($45)

**Day 3**

* **Breakfast:** Vancouver Breakfast Co. ($25)
* **Attraction 1:** Capilano Suspension Bridge ($55)
* **Attraction 2:** Stanley Park ($0)
* **Lunch:** Japadog ($20)
* **Attraction 3:** Vancouver Aquarium ($40)
* **Attraction 4:** Gastown ($0)
* **Dinner:** L'Abattoir ($50)

[END AGENT RESPONSE]

The constraint is: Each day in the itinerary must correspond to a budget of 150$.

[BEGIN EVALUATION PROCESS]
RATIONALE: We need to calculate the total cost for each day in the itinerary, which has 3 days. For day 1, the relevant items that we need to sum are: [20, 35, 30, 20, 0, 0, 45]. Next, we will calculate their sum: 20 + 35 + 30 + 20 + 0 + 0 + 45 = 150. Thus, day 1 corresponds to a 150$ budget and we can continue and check the next day. For day 2, the relevant items that we need to sum are: [20, 15, 20, 30, 30, 0, 45]. Next, we will calculate their sum: 20 + 15 + 20 + 30 + 30 + 0 + 45 = 160. Thus, day 2 does not correspond to a 150$ budget, so we do not need to check the next day. We can conclude that the agent's response does not satisfy the constraint. # [END_RATIONALE]
FINAL_ANSWER: no
[END EVALUATION PROCESS]

The agent's response is:
[BEGIN AGENT RESPONSE]
**Driving Plan from San Francisco to Las Vegas**

**Segment 1: San Francisco to Fresno**

* Take I−5 S
* Driving time: 3 hours
* Average speed: 62 mph

**Stop 1: Harris Ranch**

* Known for its restaurant and hotel
* Break for lunch or an overnight stay

**Segment 2: Fresno to Barstow**

* Continue on I−5 S to CA−99 S
* Then take CA−58 E
* Driving time: 3 hours 15 minutes
* Average speed: 60 mph

**Stop 2: Mojave National Preserve**

* Explore the scenic desert landscape
* Consider a short hike or scenic drive

**Segment 3: Barstow to Las Vegas**

* Take I−15 N
* Driving time: 2 hours 30 minutes
* Average speed: 60 mph

**Stop 3: Primm Valley Resorts**

* Located at the California−Nevada border
* Opportunity for a break or to enjoy entertainment options

**Arrival in Las Vegas**

[END AGENT RESPONSE]

The constraint is: The driving distance in each driving segment must be no more than 200 miles.

[BEGIN EVALUATION PROCESS]
RATIONALE: We need to calculate the driving distance for each segment, and there are 3 segments. For segment 1, the driving time is 3 hours, and average driving speed is 62 mph. Thus, the driving distance is 3 * 62 = 186 miles, which is less than 200 miles. Thus, we can continue checking the next segment. For segment 2, the driving time is 3 hours and 15 minutes (3.25 in decimal representation), and average driving speed is 60 mph. Thus, the driving distance is 3.25 * 60 = 195 miles, which is less than 200 miles. Thus, we can continue checking the next segment. For segment 3, the driving time is 2 hours and 30 minutes (2.5 in decimal representation), and average driving speed is 60 mph. Thus, the driving distance is 2.5 * 60 = 150 miles, which is less than 200 miles. Thus, the agent's response does satisfy the constraint. # [END_RATIONALE]
FINAL_ANSWER: yes
[END EVALUATION PROCESS]

[END  EXAMPLES]

