# OpenReview forum: "The Ability of Large Language Models to Evaluate Constraint-satisfaction in Agent Responses to Open-ended Requests"
_ICLR.cc/2025/Conference — Submitted to ICLR 2025_

### Official Review · Reviewer_gd6g · 2024-11-04

**Soundness:** 3
**Presentation:** 4
**Contribution:** 1
**Rating:** 3
**Confidence:** 3

**Summary:**

This paper studies an LLM evaluation setup called NORA (no one right answer).

Within this context, it proposes a dataset and the associated design decisions to evaluate the LLM response generation for queries in different domains, including meal planning, daily scheduling, and workout planning.

Several LLMs and prompting techniques are then evaluated in this dataset, which is contributed to the community.

**Strengths:**

First of all, I enjoyed reading this paper. The paper write-up is easy to follow, and the material is presented naturally with suitable examples in the right places. Thanks to our authors!
A non-trivial amount of work is needed to compile this dataset, which is said to have contributed to the community and can be extremely valuable.
The authors are honest about the design choices that went into this process, clear about the dataset created and utilized, and, in particular, cover several limitations of their work. The paper attains a scholarly approach.

**Weaknesses:**

I am following the introduction of the dataset, which is a non-trivial amount of work and nicely contributes to the literature and community. This is all great.
What is less clear to me is 1) the high-level motivation behind doing this, and 2) the ultimate path this might take our understanding further.

Specific to the evaluation dataset, as noted by the authors, GPT4o achieves %97+ accuracy. It is not clear how much further this can be improved (I assume there will always be some data uncertainty etc.) and what value remains. This is quite important if this dataset is to be serving as a "benchmark". This might not be the case here.
I understand that experiments rely on LLMs only but I would have liked to see a comparison with an LLM with tools access as another additional baseline. Some of the steps required such as counting, arithmetic, summation, etc. are clear candidates for tool usages. The paper claims that most errors are due to reasoning rather than calculation (which I tend to agree) but still an LLM with a tool is an immediate baseline to include in experiments. I wonder if that would open-source LLMs, which are trailing behind GPT, to close the gap.

**Questions:**

Two high-level questions;
1) What was the motivation behind studying these particular domains: meal planning, daily scheduling, and workout planning?
2) As a takeaway from this study, would you suggest that we need datasets for domain/application to advance LLMs constraint satisfaction evaluation?

---

### Official Review · Reviewer_PJXZ · 2024-11-04

**Soundness:** 2
**Presentation:** 2
**Contribution:** 3
**Rating:** 5
**Confidence:** 3

**Summary:**

This paper introduces a new dataset for QA questions with constraints. The dataset comprises four parts: user question, constraint, agent response, and the correct label for the response. It focuses on tasks related to arithmetic calculation and counting. The paper also evaluates several well-known LLMs on the new dataset and finds that GPT-4o achieves the best performance. The results suggest that the primary source of errors stems from reasoning challenges rather than arithmetic calculation abilities.

**Strengths:**

Strengths:
1. Development of the ACS dataset for benchmarking auto-scorers of constraint satisfaction.
2. Benchmark testing on various LLM models.

**Weaknesses:**

Weaknesses:
1. Some parts of the writing are unclear.
2. There are too many manual steps involved in dataset generation.
3. The dataset is relatively small.

**Questions:**

The authors could consider adding logical constraints, such as Boolean constraints, which could incorporate both numerical and logical elements.

The writing is sometimes hard to follow. The authors could provide more examples for each component. Initially, I thought the paper focused on a dataset intended for LLMs to generate correct responses for questions with constraints. However, midway through, I realized the LLMs are actually expected to assess whether the provided responses satisfy the constraints. I hope the authors can clarify this at the beginning of the paper.

- Line 105: “The dataset if available” should be "is" instead of "if".

- Line 162: This section could benefit from examples at each step.

- Line 242: There are two periods.

- Lines 268-269: It is unclear how "The maximal number of tokens from the agent response field in the ACS dataset is 1963" relates to "relevant information is scattered in the context window."

- Lines 287-289: The meaning of "keywords that represent the constraint as being satisfied" is unclear, as is the example that follows. And why is “the label unsatisfied” in the example?

There are too many uses of right” , In LaTeX, you can use `' instead.

---

### Official Review · Reviewer_pPVj · 2024-11-04

**Soundness:** 2
**Presentation:** 1
**Contribution:** 1
**Rating:** 3
**Confidence:** 5

**Summary:**

The authors present a 405 item benchmark meant to analyze the ability of LLMs to evaluate whether a numerical constraint has been satisfied in a given input. The authors generate input texts and numerical constraints using a series of LLM calls together with manual cleanup: one to generate a constraint-including request, another to generate a responses to these requests, and another to extract numerical constraints from the LLM-generated request.
Final input texts come in four varieties: meal plans, daily schedule plans, cardio workout plans, and strength workout plans. Each prompt presents the input text without any further context, and then asks the LLM to evaluate whether one specific constraint is satisfied or not.

The authors then evaluate models from the Gemini, GPT, Llama, Mixtral, and Mistral families on their benchmarks, finding that performance varies. All models score over 72%, and GPT-4o achieves an accuracy of 97% on the dataset. For a subset of these LLMs, the authors then manually categorize the errors seen in their CoT traces. The authors also propose using a variation of their benchmark generation process as a protocol for automatic evaluation of LLM responses to user queries.

**Strengths:**

1. The authors restrict their analysis to arithmetical conststraint evaluation on problems constructed to have a well-defined ground truth, sidestepping a significant problem of some previous benchmarks.
1. The paper evaluates both proprietary and open models on the dataset, providing broad coverage of different LLM families.
1. The authors provide analysis beyond just raw accuracy scores (F1, error categorization) and show performance on more than one prompt type.
1. The authors will release the dataset after the review process, which is good practice.
1. Section 2.1 provides a thorough overview of the steps involved in the dataset generation process.

**Weaknesses:**

In general, this paper would benefit from substantially more time, a clearer focus (is it aiming to be a benchmark or a validation framework?), and a more carefully constructed and relevant dataset.
In its current form, it feels far from complete, and I cannot recommend it for publication. I'll do my best to constructively break down why I believe this:

The premise of the paper is valid and asks an important question: can LLMs be useful as automatic constraint satisfaction verifiers for LLM responses to user queries?
The current direction has promise for making a useful contribution to answering this question, but only if much more is added.
1. The paper raises many immediately relevant questions but dismisses them as further work throughout the body of the work (lines 158, 352, 433, 447, 455). This "further work" includes significant amounts of analysis that I believe should actually be in the main results of this paper, especially the F1 score analysis (line 158) and examining intermediate numerical values rather than just final boolean evaluations (line 447).
1. GPT-4o succeeds on over 97% of all problems in the dataset, thus saturating it. If this were a benchmark which was a well-validated representation of the domain it claimed to examine, this could be taken as evidence that this model is already sufficient for the task of evaluating constraint satisfaction in responses to user queries. However, as the authors themselves admit, this is a deliberately simplified dataset: only one constraint is evaluated at a time, and only the simplest kinds of arithmetic constraints are represented. A more useful evaluation set would either provide significant evidence that the current state of the art is sufficient for the task or would highlight where and how SoTA currently fails, thus giving potential directions for improvement (whether train time or via inference-time augmentations).
1. The error analysis is unclear and incomplete. The biggest category by far is "other." The classification of errors is confusing: counting is considered a "reasoning" task, while multiplication (a procedure that, in a sense, generalizes counting) isn't. In general, the definition of reasoning here seems both very broad and inconsistent. Why is information extraction a reasoning task? For clarity, I would recommend removing the "reasoning" label entirely from Table 2 and its associated analysis paragraphs.
1. As the authors mention in the introduction, there have been many prior benchmarks and analyses that have looked at the same LLM abilities as those being tested here. The fact that the generation being evaluated by the LLM was in response to a "NORA" prompt doesn't fundamentally change the nature of the benchmark questions, as all of them *do* have one right answer (either "satisfiable" or "unsatisfiable"). This is especially true given that the full dataset was automatically generated rather than sourced from a real setting.
1. The dataset presented is too small and irrelevant. While the authors claim that it is "difficult" and "realistic," it is (as mentioned) already saturated, and no justification is provided for its realism. Currently it consists of a handful of very small artificial domains and uses an LLM as a user proxy but does not attempt to examine the validity of this use. A much more useful dataset would consist of actual user requests "in the wild" together with a selection of LLM generations (ideally across various models). At the very least, any benchmark purporting to answer the questions this paper is interested in should validate whether its benchmark is reflective of the sorts of queries and constraints LLMs are likely to encounter in practice.
1. The paper's focus is also unclear. The introduction mentions an automatic evaluation framework for LLMs tasked with constrained query generation and presents a figure, but this framework is not mentioned in the body of the paper. Multiple pieces of this system are not described in detail: What is the scoring system? What is the intended role of this system? For example, is the generation model reprompted if the evaluation comes up negative (or just has too low a score? in which case how is the cutoff calculated?)? If the paper is meant to be a benchmark paper, then it is unclear what the purpose of describing the framework is, unless it is modified and expanded on to create something like a living benchmark which grows with more queries or similar. If the framework is meant as a standalone contribution, then the paper must include not only further description but also further analysis of it, as well as evaluations of whether and how much it improves performance.

## Nitpicks/Typos
1. The paper introduces a number of new (nonstandard) acronyms. In particular, WOV is used only three more times after it is introduced. This makes the paper harder to read. It would be clearer if these terms were spelled out each time.
1. Line 52: while the proposed benchmark does seem to have well-defined constraints (I haven't seen the prompts), the use case it purports to model often will not. Non-arithmetical constraints which may seem at first blush to be objective and well-defined may not fully be. The example the authors give "a 3-day meal-plan with no meat products" arguably has this property. Must a 3-day meal plan include three meals a day? Does fasting for three days satisfy the requirements? Which definition of meat are we using? Does fish count? Do insects? Does lab-grown meat count? E.g. the state of Missouri required "meat" to be derived from "harvested production livestock or poultry," which would answer no to all three of the meat questions. In general, polysemanticity and linguistic ambiguity makes it very difficult to make strong claims about whether constraints are truly well-defined. If the authors wish to retain this aspect of their analysis, I think it is important that they discuss these issues, and whether and to what extent well-defined, objective, and verifiable criteria exist beyond mathematical tasks.
1. Line 105: "The dataset **if** available at *blinded for submission*." (Typo)
1. Line 167-168: "generate the text in all stages decribes next:" (Typo)
1. Line 298: The authors claim that the problems "can be seen" as "elementary school level," but provide no justification for why this is true or why it matters. It is well-known that LLMs fail on simple grade school tasks like multiplication (even with CoT/scratch space), but can score highly on some graduate level exams. Evaluations and rankings based on approximately equivalent human school level do not seem particularly meaningful for these systems.
1. Line 432: "composed of very large tokens" should be "composed of very large numbers of tokens"

**Questions:**

1. Who were your human raters/annotators? I'd appreciate more detail on that aspect of dataset construction (and how/whether these annotations were validated e.g. via inter-annotator agreements).
1. Could you include examples of the actual prompts across all four domains? In particular, is the user query included in these? It doesn't seem to be in the (out of domain) few-shot examples shown in the appendix.
1. In step 6 of section 2.1, you mention the possibility of using other models to generate responses for this dataset. Did you compare generations across models? I'd be especially curious to know if some are more or less like real human requests or if they are easier or harder for different models to evaluate.
1. Could you provide examples of each error type? I didn't see them in the main text or appendix, making it hard to understand what exactly the error types mean.
1. Line 74: What do you mean when you say that using human raters often isn't reproducible? I would have assumed that a failure to reproduce human ratings with other humans would mean that there is disagreement about the ground truth of the claim (and thus the constraint wasn't well-defined or objective), but I think I must be misunderstanding what the sentence is trying to say.

---

### Official Review · Reviewer_8RfG · 2024-11-05

**Soundness:** 2
**Presentation:** 3
**Contribution:** 2
**Rating:** 5
**Confidence:** 2

**Summary:**

The paper presents an evaluation framework for generative AI agents (LLM based agents) for No-One-Right-Answer (NORA) scenarios. Specifically, the focus of the paper is on developing a benchmark dataset called Arithmetic Constraint Satisfaction (ACS) dataset for evaluating if certain arithmetic constraints are satisfied by the agent's response. In particular, verifying that such a constraint is satisfied requires basic operations such as summation, subtraction, multiplication and counting, as well as the entire generated response instead of just specific parts of it. The empirical evaluation is carried out using both closed as well as open models, including OpenAI's GPT models, Google's Gemini models as well as Meta's Llama models, respectively. The results demonstrate the capability of modern language models to generate NORA responses that indeed satisfy the user specified constraints.

**Strengths:**

- Evaluating if the LLM's responses satisfy the user specified constraints is important in many realistic deployments of LLLM based AI agents. Therefore, developing a benchmark dataset for these kinds of situations is definitely called for.

- The quality of the presentation is overall quite good and therefore the paper is relatively easy to follow even by readers outside this research area. Most of the technical details presented in the paper are discussed in a relatively clear manner.

**Weaknesses:**

- Most modern LLMs do fairly well on arithmetic reasoning problems and the results in table 1 aren't actually surprising. Therefore, I think the proposed ACS dataset may be too easy for constraint satisfaction task considered.

**Questions:**

- It would be interesting to verify if implicit constraints are satisfied. For example in the event planning scenario, ensuring that the agent doesn't schedule an event in two different places at the same time. Could you comment a bit on how the ACS dataset could be extended to address these scenarios.

---

### Meta-Review · Area_Chair_xHh7 · 2024-12-16

**Metareview:**

Reviewers agreed that datasets and evaluations of LLMs are important, yet also raised various concerns on the approaches of this paper, including conducting more immediately relevant questions, providing more complete and clear analysis, and improving the datasets in quantity and quality. The authors did not provide a response so the scores and sentiments remained negative after the rebuttal phase.

**Additional Comments On Reviewer Discussion:**

See above

---

### Decision · Program_Chairs · 2025-01-22

Reject